# Revisiting Automated Topic Model Evaluation with Large Language Models

**Dominik Stammbach**[E]        **Vilém Zouhar**[E]        **Alexander Hoyle**[M]
**Mrinmaya Sachan**[E]        **Elliott Ash**[E]
[E]ETH Zürich        [M]University of Maryland
{dominsta,vzouhar,ashe,msachan}@ethz.ch    hoyle@umd.edu

## Abstract

Topic models help make sense of large text collections. Automatically evaluating their output and determining the optimal number of topics are both longstanding challenges, with no effective automated solutions to date. This paper evaluates the effectiveness of large language models (LLMs) for these tasks. We find that LLMs appropriately assess the resulting topics, correlating more strongly with human judgments than existing automated metrics. However, the type of evaluation task matters — LLMs correlate better with coherence ratings of word sets than on a word intrusion task. We find that LLMs can also guide users toward a reasonable number of topics. In actual applications, topic models are typically used to answer a research question related to a collection of texts. We can incorporate this research question in the prompt to the LLM, which helps estimate the optimal number of topics.

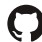 github.com/dominiksinsaarland/
evaluating-topic-model-output

## 1 Introduction

Topic models are, loosely put, an unsupervised dimensionality reduction technique that help organize document collections (Blei et al., 2003). A topic model summarizes a document collection with a small number of *topics*. A topic is a probability distribution over words or phrases. A topic $T$ is interpretable through a representative set of words or phrases defining the topic, denoted $W_T$.[1]

Each document can, in turn, be represented as a distribution over topics. For each topic, we can retrieve a representative document collection by sorting documents across topic distributions. We denote this set of documents for topic $T$ as $D_T$. Because of their ability to organize large collections of texts, topic models are widely used

in the social sciences, digital humanities, and other disciplines to analyze large corpora (Talley et al., 2011; Grimmer and Stewart, 2013; Antoniak et al., 2019; Karami et al., 2020, inter alia).

Interpretability makes topic models useful, but human interpretation is complex and notoriously difficult to approximate (Lipton, 2018). Automated topic coherence metrics do not correlate well with human judgments, often overstating differences between models (Hoyle et al., 2021; Doogan and Buntine, 2021). Without the guidance of an automated metric, the number of topics, an important hyperparameter, is usually derived manually: Practitioners fit various topic models, inspect the resulting topics, and select the configuration which works best for the intended use case (Hoyle et al., 2021). This is a non-replicable and time-consuming process, requiring expensive expert labor.

Recent NLP research explores whether large language models (LLMs) can perform automatic annotations; e.g., to assess text quality (Fu et al., 2023; Faggioli et al., 2023; Huang et al., 2023, inter alia). Here, we investigate whether LLMs can automatically assess the coherence of topic modeling output and conclude that:

> (1) LLMs can accurately judge topic coherence,
> (2) LLMs can assist in automatically determining reasonable numbers of topics.

We use LLMs for two established topic coherence evaluation tasks and find that their judgment strongly correlates with humans on one of these tasks. Similar to recent findings, we find that coherent topic word sets $W_T$ do not necessarily imply an optimal categorization of the document collection (Doogan and Buntine, 2021). Instead, we automatically assign a label to each document in a $D_T$ and choose the configuration with the purest assigned labels. This solution correlates well with an underlying ground truth. Thus, LLMs can help find good numbers of topics for a text collection, as we show in three case studies.

---

[1]We think of "words" as an atomic unit in a document, which can also be an n-gram or phrase. E.g., $W_{\text{legal}}$ = {litigation, attorney-client privilege, intellectual property, ... }.

## 2 Topic Model Evaluation

Most topic model evaluations focus on the *coherence* of $W_T$, the most probable words from the topic-word distribution (Röder et al., 2015). Coherence itself can be thought of as whether the top words elicit a distinct concept in the reader (Hoyle et al., 2021). To complicate matters, human evaluation of topic models can be done in diverse ways. E.g., we can ask humans to directly rate topic coherence, for example, on a 1-3 scale (Newman et al., 2010a; Mimno et al., 2011; Aletras and Stevenson, 2013, inter alia). We can also add an unrelated intruder word to the list of top words, which human annotators are asked to identify. The intuition is that intruder words are easily identified within coherent and self-contained topics, but hard to identify for incoherent or not self-contained topics (Chang et al., 2009). High human accuracy on this task is thus a good proxy for high topic coherence. See both in Example 1.

**Intrusion Detection Task**

| water | area | river | park | miles | game |
|-------|------|-------|------|-------|------|
| horses | horse | breed | hindu | coins | silver |

**Rating Task**

| health | hospital | medicare | welfare | insure | 3 |
|--------|----------|----------|---------|--------|---|
| horses | zurich | race | | dog | canal | 1 |

Example 1: Two examples of intrusion detection (select outlier) and topic rating tasks (rate overall coherence). Each example is on separate row.

Although many automated metrics exist (Wallach et al., 2009; Newman et al., 2010b; Mimno et al., 2011; Aletras and Stevenson, 2013), normalized pointwise mutual information (NPMI, Bouma, 2009) is the most prevalent when evaluating novel methods (Hoyle et al., 2021). Informally, NPMI is larger if two words co-occur together regularly in a reference corpus. Another popular metric, $C_v$, is a combination of NPMI and other measures and is also popular (Röder et al., 2015). See the formula definitions in Appendix D.

Despite their popular use, these metrics correlate poorly with human evaluations (Hoyle et al., 2021; Doogan and Buntine, 2021). In this work, we let LLMs perform the rating and intrusion detection tasks for topic model evaluation[2] and propose LLM scores as a novel automated metric. Similar

---

[2] We use ChatGPT as the main LLM (chat.openai.com). We list ablation results using other LLMs in Appendix B.

work by Rahimi et al. (2023) is carried contemporaneously. LLMs have already been used to rank machine translations and generated text (Zhang et al., 2020; Fu et al., 2023; Kocmi and Federmann, 2023) and have also been shown to perform on par with crowdworkers (Gilardi et al., 2023).

| Task | Dataset | NPMI | $C_v$ | LLM | Ceiling |
|------|---------|------|-------|-----|---------|
| Intrusion | NYT | 0.43 | 0.45† | 0.37 | 0.67 |
| | Wiki | 0.39† | 0.34 | 0.35 | 0.60 |
| | Both | 0.40† | 0.40† | 0.36 | 0.64 |
| Rating | NYT | 0.48 | 0.40 | 0.64⋆ | 0.72 |
| | Wiki | 0.44 | 0.40 | 0.57⋆ | 0.56 |
| | Both | 0.44 | 0.42 | 0.59⋆ | 0.65 |

Table 1: Spearman correlation between human scores and automated metrics. All results use 1000 bootstrapping episodes — re-sampling human annotations and LLM scores, and averaging the correlations. Marked ⋆ if significantly better than second best (<0.05), otherwise †. **Ceiling** shows batched inter-annotator agreement.

## 3 LLM and Coherence

First, we show that large language models can assess the quality of topics generated by different topic modeling algorithms. We use existing topic modeling output annotated by humans (Hoyle et al., 2021).[3] This data consists of 300 topics, produced by three different topic modeling algorithms on two datasets: NYtimes (Sandhaus, 2008) and Wikitext (Merity et al., 2017). For each of the 300 topics, there are 15 individual human annotations for the topic word relatedness (on 1-3 scale), and 26 individual annotations for whether a crowd-worker correctly detected an intruder word. We replicate both tasks, prompting LLMs instead of human annotators. See Prompt 1 for prompt excerpts, and Appendix A for full details.

**Intruder detection prompt.**
**System prompt:** [...] Select which word is the least related to all other words. If multiple words do not fit, choose the word that is most out of place. [...]
**User prompt:** water, area, river, park, miles, game

**Rating Task prompt.**
**System prompt:** [...] Please rate how related the following words are to each other on a scale from 1 to 3 ("1" = not very related, "2" = moderately related, "3" = very related). [...]
**User prompt:** lake, park, river, land, years, feet, ice, miles, water, area

Prompt 1: LLM prompts for assessing topic coherence.

We compute the Spearman correlation between the LLM answer and the human assessment of the topics and show results in Table 1.

---

[3] Models: Gibbs-LDA (McCallum, 2002), Dirichlet-VAE (Burkhardt and Kramer, 2019), and ETM (Dieng et al., 2020).

**Baseline metrics.** For NPMI and $\mathbf{C_v}$, we report the best correlation by Hoyle et al. (2021). These metrics depend on the reference corpus and other hyperparameters and we always report the best value. Hoyle et al. (2021) find no single *best* setting for these automated metrics and therefore this comparison makes the baseline inadequately strong.

**Intrusion detection task.** The accuracies for detecting intruder words in the evaluated topics are almost identical – humans correctly detect 71.2% of the intruder words, LLMs identify intruders in 72.2% of the cases. However, humans and LLMs differ for which topics these intruder words are identified. This results in overall strong correlations within human judgement, but not higher correlations than NPMI and $\mathbf{C_v}$ (in their best setting).

**Coherence rating task.** The LLM rating of the $W_T$ top word coherence correlates more strongly with human evaluations than all other automated metrics in any setting. This difference is statistically significant, and the correlation between LLM ratings and human assessment approaches the inter-annotator agreement ceiling. Appendix Appendix B shows additional results with different prompts and LLMs.

**Recommendation.** Both findings support using LLMs for evaluating coherence of $W_T$ in practice as they correlate highly with human judgements.

## 4 Determining the Number of Topics

Topic models require specifying the number of topics. Practitioners usually run models multiple times with different numbers of topics (denoted by $k$). After manual inspection, the model which seems most suited for a research question is chosen. Doogan et al. (2023) review 189 articles about topic modeling and find that common use cases are exploratory and descriptive studies for which no single best number of topics exists. However, the most prevalent use case is to isolate semantically similar documents belonging to topics of interest. For this, Doogan and Buntine (2021) challenge the focus on only evaluating $W_T$, and suggest an analysis of $D_T$ as well. If we are interested in organizing a collection, then we would expect the top documents in $D_T$ to receive the same topic labels. We provide an LLM-based strategy to determine good number of topics for this use case: We let an LLM assign labels to documents, and find that topic assignments with greater label purity correlate with the ground-truth in three case studies.

Topics of interest might be a few broad topics such as *politics* or *healthcare*, or many specific topics, like *municipal elections* and *maternity care*. Following recent efforts that use research questions to guide LLM-based text analysis (Zhong et al., 2023), we incorporate this desideratum in the LLM prompt. We run collapsed Gibbs-sampled LDA (in MALLET: McCallum, 2002) on two text collections, with different numbers of topics ($k = 20$ to 400), yielding 20 models per collection. To compare topic model estimates and ground-truth partitions, we experiment with a legislative Bill summary dataset (from Hoyle et al., 2022) and Wikitext (Merity et al., 2017), both annotated with ground-truth topic labels in different granularities.

### 4.1 Proposed Metrics

**Ratings algorithm.** For each of the 20 models, we randomly sample $W_T$ for some topics and let the LLM rate these $W_T$. The prompt is similar to the ratings prompt shown in Prompt 1, see Appendix E for full details. We then average ratings for each configuration. Intuitively, the model yielding the most coherent $W_T$ should be the one with the optimal topic count. However, this procedure does not correlate with ground-truth labels.

**Text labeling algorithm.** Doogan and Buntine (2021) propose that domain experts assign labels to each document in a $D_T$ instead. A good topic should have a coherent $D_T$: The same label assigned to most documents. Hence, good configurations have high purity of assigned labels within each topic. We proceed analogously. For each of the 20 models, we randomly sample $D_T$ for various topics. We retrieve the 10 most probable documents and then use the LLM to assign a label to these documents. We use the system prompt *[...] Annotate the document with a broad\narrow label [...]*, see Appendix E for full details. We compute the purity of the assigned labels and average purities and we select the configuration with the most pure topics. In both procedures, we smooth the LLM outputs using a rolling window average to reduce noise (the final average goodness is computed as moving average of window of size 3).

### 4.2 Evaluation

We need a human-derived metric to compare with the purity metric proposed above. We measure

Figure 1: (1) ARI for topic assignment and ground-truth topic labels, (2) LLM word set coherence, (3) LLM document set purity, obtained by our algorithm. ARI correlates with LLM document set purity, but not with LLM word set coherence. The ground-truth number of topics are: 21 topics in the BillSum dataset, 45 broad topics in Wikitext and 279 specific topics in Wikitext. $\rho_D$ and $\rho_W$ are document-LLM and word-LLM correlations with ARI.

the alignment between a topic model's predicted topic assignments and the ground-truth labels for a document collection (Hoyle et al., 2022).

We choose the *Adjusted Rand Index (ARI)* which compares two clusterings (Hubert and Arabie, 1985) and is high when there is strong overlap. The predicted topic assignment for each document is its most probable topic. Recall that there exist many different optimal topic models for a single collection. If we want topics to contain semantically similar documents, each ground-truth assignment reflects one possible set of topics of interests.

If our LLM-guided procedure and the ARI correlate, this indicates that we discovered a reasonable value for the number of topics. In our case, the various ground-truth labels are assigned with different research questions in mind. We incorporate such constraints in the LLM prompt: We specify whether we are interested in broad or specific topics, and we enumerate some example ground-truth categories in our prompt. Practitioners usually have priors about topics of interest before running topic models, thus we believe this setup to be realistic.

In Figure 1 we show LLM scores and ARI for broad topics in the Bills dataset. We used this dataset to find a suitable prompt, hence this could be considered the "training set". We plot coherence ratings of word sets in blue, purity of document labels in red, and the ARI between topic model and ground-truth assignments in green. The purity of LLM-assigned $D_T$ labels correlate with the ARI, whereas the $W_T$ coherence scores do not. The argmax of the purity-based approach leads to similar numbers of topics as suggested by the ARI argmax (although not always the same).

For Wikitext, we evaluate the same 20 topic models, but measure ARI between topic model assignment and two different ground-truth label sets. The LLM scores differ only because of different prompting strategies. The distributions indicate that this

strategy incorporates different research questions.

For Bills, our rating algorithm suggests to use a topic model with $k=100$ topics. In Appendix G, we show corresponding word sets. The resulting $W_T$ seem interpretable, although the ground-truth assignments using document-topic estimates are not correlated with the ground-truth labels. The purity-based approach instead suggests to use $k=20$ topics, the same $k$ as indicated by the ARI. We show ground-truth labels and LLM-obtained text labels in Appendix G. We further manually evaluate 180 assigned LLM-labels and find that 94% of these labels are reasonable. Appendix F shows further evaluation of these label assignments.

## 5 Discussion

In this work, we revisit automated topic model evaluation with the help of large language models. Many automated evaluation metrics for topic models exist, however these metrics seem to not correlate strongly with human judgment on word-set analysis (Hoyle et al., 2021). Instead, we find that an LLM-based metric of coherent topic words correlates with human preferences, outperforming other metrics on the rating task.

Second, the number of topics $k$ has to be defined before running a topic model, so practitioners run multiple models with different $k$. We investigate whether LLMs can guide us towards reasonable $k$ for a collection and research question. We first note that the term *optimal number of topics* is vague and that such quantity does not exist without additional context. If our goal is to find a configuration which would result in coherent document sets for topics, our study supports evaluating $D_T$ instead of $W_T$, as this correlates more strongly with the overlap between topic model and ground-truth assignment. This finding supports arguments made in Doogan and Buntine (2021) who challenge the focus on $W_T$ in topic model evaluation.

## Limitations

**Choice of LLM.** Apart from ChatGPT, we also used open-source LLMs, such as FLAN-T5 (Chung et al., 2022), and still obtained reasonable, albeit worse than ChatGPT, coherence correlations. Given the rapid advances, future iterations of open-source LLMs will likely become better at this task.

**Number of topics.** The *optimal* number of topics is a vague concept, dependent on a practitioner's goals and the data under study. At the same time, it is a required hyperparameter of topic models. Based on Doogan et al. (2023), we use an existing document categorization as *one possible* ground truth. While content analysis is the most popular application of topic models (Hoyle et al., 2022), it remains an open question how they compare to alternative clustering algorithms for this use case (e.g., k-means over document embeddings).

**Interpretability.** LLM label assignment and intruder detection remain opaque. This hinders the understanding of the evaluation decisions.

**Topic modeling algorithm.** In Section 3, we evaluate three topic modeling algorithms: Gibbs-LDA, Dirichlet-VAE and ETM (see Hoyle et al., 2021). In Section 4, we use only Gibbs-LDA and expansion to further models is left for future work.

**Future work.**
- Evaluation of clustering algorithms with LLMs (e.g., k-means).
- More rigorous evaluation of open-source LLMs.
- Formalization, implementation and release of an LLM-guided algorithm for automatically finding optimal numbers of topics for a text collection and a research question.

## Ethics Statement

**Using blackbox models in NLP.** Statistically significant positive results are a sufficient proof of models' capabilities, assuming that the training data is not part of the training set. This data leakage problem with closed-source LLMs is part of a bigger and unresolved discussion. In our case, we believe data leakage is unlikely. Admittedly, the data used for our coherence experiments has been publicly available. However, the data is available in a large JSON file where the topic words and annotated labels are stored disjointly. For our case studies in Section 4, the topic modeling was constructed as part of this work and there is no ground-truth which could leak to the language model.

**Negative results with LLMs.** In case of negative results, we cannot conclude that a model can not be used for a particular task. The negative results can be caused by inadequate prompting strategies and may even be resolved by advances in LLMs.

**LLMs and biases.** LLMs are known to be biased (Abid et al., 2021; Lucy and Bamman, 2021) and their usage in this application may potentially perpetuate these biases.

**Data privacy.** All data used in this study has been collected as part of other work. We find no potential violations of data privacy. Thus, we feel comfortable re-using the data in this work.

**Misuse potential.** We urge practicioners to not blindly apply our method on their topic modeling output, but still manually validate that the topic outputs would be suitable to answer a given research question.

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

## A  Language Model Prompts

In this section, we show the used LLM prompts. The task descriptions are borrowed from (Hoyle et al., 2021) and mimic crowd-worker instructions. We use a temperature of 1 for LLMs, and the topic words are shuffled before being prompted. Both introduce additional variation within the results, similar to how some variation is introduced if different crowd-workers are asked to perform the same task.

**Intruder detection task.**  Analogous to the human experiment, we randomly sample (a) five word from the top 10 topic words and (b) an additional intruder word from a different topic which does not occur in the top 50 words of the current topic. We then shuffle these six words. We show the final prompt with an example topic in Prompt 2. We also construct a prompt without the dataset description (see Prompt 3 and results in Table 2).

**System prompt:** You are a helpful assistant evaluating the top words of a topic model output for a given topic. Select which word is the least related to all other words. If multiple words do not fit, choose the word that is most out of place. The topic modeling is based on The New York Times corpus. The corpus consists of articles from 1987 to 2007. Sections from a typical paper include International, National, New York Regional, Business, Technology, and Sports news; features on topics such as Dining, Movies, Travel, and Fashion; there are also obituaries and opinion pieces. Reply with a single word.
**User prompt:** water, area, river, park, miles, game

Prompt 2: Intruder Detection Task (the intruder word in this topic is *game*). We show the task description for the New York Times dataset in the prompt for the rating task (the dataset descriptions are kept the same).

**System prompt:** You are a helpful assistant evaluating the top words of a topic model output for a given topic. Select which word is the least related to all other words. If multiple words do not fit, choose the word that is most out of place. Reply with a single word.
**User prompt:** water, area, river, park, miles, game

Prompt 3: Intruder Detection Task. The intruder word in this topic is *game*.

**Rating Task.**  Similar to the human experiment, we retrieve the top 10 topic words and shuffle them.

We include a task and dataset description which leads to Prompt 4. The minimal prompt without the dataset description is shown in Prompt 5.

**System prompt:** You are a helpful assistant evaluating the top words of a topic model output for a given topic. Please rate how related the following words are to each other on a scale from 1 to 3 ("1" = not very related, "2" = moderately related, "3" = very related).
The topic modeling is based on the Wikipedia corpus. Wikipedia is an online encyclopedia covering a huge range of topics. Articles can include biographies ("George Washington"), scientific phenomena ("Solar Eclipse"), art pieces ("La Danse"), music ("Amazing Grace"), transportation ("U.S. Route 131"), sports ("1952 winter olympics"), historical events or periods ("Tang Dynasty"), media and pop culture ("The Simpsons Movie"), places ("Yosemite National Park"), plants and animals ("koala"), and warfare ("USS Nevada (BB-36)"), among others. Reply with a single number, indicating the overall appropriateness of the topic.
**User prompt:** lake, park, river, land, years, feet, ice, miles, water, area

Prompt 4: Rating Task. Topic terms are shuffled.

**System prompt:** You are a helpful assistant evaluating the top words of a topic model output for a given topic. Please rate how related the following words are to each other on a scale from 1 to 3 ("1" = not very related, "2" = moderately related, "3" = very related).
Reply with a single number, indicating the overall appropriateness of the topic.
**User prompt:** lake, park, river, land, years, feet, ice, miles, water, area

Prompt 5: Rating Task without dataset description. Topic terms are shuffled.

## B  Additional: Topic Model Outputs

**Minimal prompt.**  Even without the dataset description in the prompt, the results remain similar.

**All human ratings.**  In our main results, we discard human annotations with low annotator confidence in the rating. We now consider all ratings, even the non-confident ones. The results are slightly better than with the filtering.

**Different LLM.**  We also evaluate both tasks with FLAN-T5 XL (Chung et al., 2022), which is instruction-finetuned across a range of tasks. This model performs well in zero-shot setting, and compares to recent state-of-the-art (Chia et al., 2023). Although it does not reach ChatGPT, the correlation with human annotators are all statistically

| Task | Dataset | NPMI | $C_v$ | LLM (main) | LLM (min.) | LLM (all ann.) | FLAN-T5 | Ceiling |
|------|---------|------|-------|------------|------------|----------------|---------|---------|
| | NYT | 0.43 | 0.45 | 0.37 | 0.41 | 0.39 | 0.37 | 0.67 |
| Intrusion | Wiki | 0.39 | 0.34 | 0.35 | 0.27 | 0.36 | 0.18 | 0.60 |
| | Both | 0.40 | 0.40 | 0.36 | 0.34 | 0.38 | 0.28 | 0.64 |
| | NYT | 0.48 | 0.40 | 0.64 | 0.64 | 0.65 | 0.31 | 0.72 |
| Rating | Wiki | 0.44 | 0.40 | 0.57 | 0.51 | 0.56 | 0.17 | 0.56 |
| | Both | 0.44 | 0.42 | 0.59 | 0.57 | 0.61 | 0.25 | 0.65 |

Table 2: Additional experiments reporting Spearman correlation between mean human scores and automated metrics. **LLM (main)** repeats our main results in Table 1 for reference. **LLM (min.)** – results using a minimal prompt without dataset descriptions. **LLM (all ann)** – no discarding low-confidence annotations. **FLAN-T5** – FLAN-T5 XL instead of ChatGPT. All numbers are the average result of 1000 bootstrapping episodes – re-sampling human annotations and LLM scores. **Ceiling** shows batched inter-annotator agreement.

significant. For the NYT and concatenated experiments, the resulting correlation are statistically indistinguishable from the best reported automated metrics NPMI and $C_v$ in (Hoyle et al., 2021). We also ran our experiments with Alpaca-7B and Falcon-7B, with largely negative results.

## C  Alternative Clustering Metrics

In our main results, we show correlations between LLM scores and the adjusted Rand Index, ARI, which measures the overlap between ground-truth clustering and topic model assignments. There are other cluster metrics, such as Adjusted Mutual Information, AMI (Vinh et al., 2010), completeness, or homogeneity. In Table 3, we show Spearman correlation statistics for these metrics. Our correlations are robust to the choice of metric used to measure the fit between the topic model assignment and the ground-truths in our case studies.

## D  Definitions

See Bouma (2009) for justification of the NPMI formula. $p(w_i)$ and $p(w_i, w_j)$ are unigram and joint probabilities, respectively.

$$\text{NPMI}(w_i, w_j) = \frac{\text{PMI}(w_i, w_j)}{-\log p(w_i, w_j)} = \frac{\log \frac{p(w_i, w_j)}{p(w_i)p(w_j)}}{-\log p(w_i, w_j)}$$

The $C_v$ metric (Röder et al., 2015) is a more complex and includes, among others, the combination of NPMI and cosine similarity for top words.

| Dataset | Topics | ARI | AMI | Compl. | Homog. |
|---------|--------|-----|-----|--------|--------|
| Bills Words | Broad | 0.61 | 0.74 | 0.63 | -0.58 |
| Wiki Words | Broad | -0.38 | -0.38 | -0.38 | 0.38 |
| Wiki Words | Specific | 0.03 | -0.24 | -0.19 | 0.17 |
| Bills Docs | Broad | 0.59 | 0.36 | 0.57 | -0.58 |
| Wiki Docs | Broad | 0.72 | 0.72 | 0.72 | -0.70 |
| Wiki Docs | Specific | 0.72 | 0.66 | 0.20 | -0.20 |

Table 3: Spearman correlation coefficients between our language-model based scores and various popular metrics for assessing the overlap between the topic model assignment and the underlying ground-truth. **Compl.** = Completeness, **Homog.** = Homogenity.

## E  Optimal Number of Topics Prompts

We now show the prompts for the optimal number of topics. We incorporate research questions in two ways: (1) we specify whether we are looking for *broad* or *narrow* topics, and (2) we prompt 5 example categories. We believe this is a realistic operationalization. If our goal is a reasonable partitioning of a collection, we usually have some priors about what categories we want the collection to be partitioned into.

Prompt 6 shows the prompt for rating $T_{ws}$ by models run with different numbers of topics. The task description and user prompt is identical to the prompt used in our prior experiments, displayed in e.g., Prompt 4. However, the dataset description is different and allows for some variation. In Prompt 7, we show the prompt for automatically assigning labels to a document from a $T_{dc}$. To automatically find the optimal number of topics for a topic model, we prompt an LLM to provide a concise label to a document from the topic document collection, the most likely documents assigned by a topic model to a topic (see Prompt 7).

You are a helpful assistant evaluating the top words of a topic model output for a given topic. Please rate how related the following words are to each other on a scale from 1 to 3 ("1" = not very related, "2" = moderately related, "3" = very related). The topic modeling is based on a legislative Bill summary dataset. We are interested in coherent *broad\narrow* topics. Typical topics in the dataset include "topic 1", "topic 2", "topic 3", "topic 4" and "topic 5". Reply with a single number, indicating the overall appropriateness of the topic. **User prompt:** lake, park, river, land, years, feet, ice, miles, water, area

Prompt 6: Rating Task without dataset description. Topic terms are shuffled. We apply this prompt to 2 different datasets and 2 different research goals (broad and narrow topics), and would set this part of the prompt accordingly. Also, we set as *topic 1* to *topic 5* the 5 most prevalent ground-truth labels from a dataset.

**System prompt:** You are a helpful research assistant with lots of knowledge about topic models. You are given a document assigned to a topic by a topic model. Annotate the document with a *broad\narrow* label, for example "topic 1", "topic 2", "topic 3", "topic 4" and "topic 5".
Reply with a single word or phrase, indicating the label of the document. **User prompt:** National Black Clergy for the Elimination of HIV/AIDS Act of 2011 - Authorizes the Director of the Office of Minority Health of the Department of Health and Human Services (HHS) to make grants to public health agencies and faith-based organizations to conduct HIV/AIDS prevention, testing, and related outreach activities ...

Prompt 7: Assigning a label to a document belonging to the top document collection of a topic. The label provided in this example is health. We apply this prompt to 2 different datasets and 2 different research goals (broad and narrow topics), and would set this part of the prompt accordingly. Also, we set as *topic 1* to *topic 5* the 5 most prevalent ground-truth labels from a dataset.

## F  Additional: Document Labeling

In our study, we automatically label the top 10 documents for five randomly sampled topics. The ARI between-topic model partitioning and ground-truth labels correlates if we were to only examine these top 10 documents or *all* documents in the collection. The correlation between these two in the Bills dataset is 0.96, indicating that analyzing only the top 10 documents in a topic is a decent proxy for the whole collection.

Next, we evaluate the LLM-based label assignement to a document. Our documents are usually long, up to 2000 words. We only consider the first 50 words in a document as input to the LLM. For Wikipedia, this is reasonable, because the first 2-3 sentences define the article and give a good summary of the topic of an article. For Bills, we manually confirm that the topic of an article is introduced at the beginning of a document.

**Human evaluation.** From each case study, we randomly sample 60 documents and assigned labels (3 examples for each of the twenty topic models), resulting in 180 examples in total. We then evaluate whether the assigned label reasonably captures the document content given the specification in the input prompt (e.g., a broad label such as *health* or *defense*, or a narrow label such as *warships of germany* or *tropical cyclones: atlantic*. Recall that the prompted labels correspond to the five most prevalent ground-truth categories of the ground-truth annotation. We find that the assigned label makes sense in 93.9% of examined labels. In the 11 errors spotted, the assigned label does not meet the granularity in 6 cases, is no adequate description of the document in 3 cases, and is a summary of the document instead of a label in 2 cases.

**Automated Metrics.** Given that we have ground-truth labels for each document, we can compute cluster metrics between the assigned labels by the LLM and the ground-truth labels (see Table 4). These values refer to comparing all labels assigned during our case study to their ground-truth label (1000 assigned datapoints per dataset).

| Dataset | Ground-Truth Labels | ARI | AMI |
|---|---|---|---|
| Bills | Broad | 19 | 43 |
| Wiki | Broad | 52 | 57 |
| Wiki | Narrow | 49 | 34 |

Table 4: Accuracy of the label assignment task. We find that the assigned labels clustering overlaps with the ground-truth labels.

On average, we assign 10 times as many unique labels to documents than there are ground-truth labels (we assign 172 different labels in the Bills dataset, 348 labels in the broad Wikitext dataset and 515 labels in the narrow Wikitext dataset). Nevertheless, the automated metrics indicate a decent overlap between ground-truth and assigned labels. Thus, the LLM often assigns the same label to documents with the same ground-truth label.

## G  Qualitative Results

In this section, we show qualitative results of our automated investigation of numbers of topics. In Table 5, we show three randomly sampled topics from the preferred topic model in our experiments. We contrast these with three randomly sampled topics from the topic model configuration which our procedures indicate as least suitable.

In Table 6, we show true labels and LLM-assigned labels for three randomly sampled topics from the preferred topic model, contrasting it with true and LLM-assigned labels from topics in the least suitable configuration. We find that indeed, the assigned labels and the ground-truth label often match – and that the purity of the LLM-assigned labels reflects the purity of the ground-truth label.

| Bills (broad categories) | |
|---|---|
| Most suitable | - veterans, secretary, veteran, assistance, service, disability, benefits, educational, compensation, veterans_affairs (3)
- land, forest, management, lands, act, usda, projects, secretary, restoration, federal (3)
- mental, health, services, treatment, abuse, programs, substance, grants, prevention, program (3) |
| Least suitable | - gas, secretary, lease, oil, leasing, act, way, federal, production, environmental (2)
- covered, criminal, history, act, restitution, child, background, amends, checks, victim (2)
- information, beneficial, value, study, ownership, united_states, act, area, secretary, new_york (1) |
| **Wikitext (broad categories)** | |
| Most suitable | - episode, star, trek, enterprise, series, season, crew, generation, ship, episodes (3)
- series, episodes, season, episode, television, cast, production, second, viewers, pilot (3)
- car, vehicle, vehicles, engine, model, models, production, cars, design, rear (3) |
| Least suitable | - episode, series, doctor, season, character, time, star, story, trek, set (2)
- stage, tour, ride, park, concert, dance, train, coaster, new, roller (1)
- said, like, character, time, life, love, relationship, later, people, way (1) |
| **Wikitext (specific categories)** | |
| Most suitable | - episode, star, trek, enterprise, series, season, crew, generation, ship, episodes (3)
- car, vehicle, vehicles, engine, model, models, production, cars, design, rear (3)
- world, record, meter, time, won, freestyle, gold, championships, relay, seconds (2) |
| Least suitable | - fossil, fossils, found, specimens, years, evolution, modern, million, eddie, like (2)
- match, event, impact, joe, team, angle, episode, styles, championship, tag (1)
- brown, rihanna, usher, love, girl, loud, yeah, wrote, bow, bad (1) |

Table 5: Most and least suitable topics according to our LLM-based assessment on different datasets and use cases. In brackets the LLM rating for the coherence of this topic.

| | Bills | | Wikitext (broad) | | Wikitext (specific) | |
|---|---|---|---|---|---|---|
| | **LLM-label** | **True label** | **LLM-label** | **True label** | **LLM-label** | **True label** |
| Most suitable | health | Health | amusement park ride | Recreation | politician | Historical figures: politicians |
| | elder abuse prevention | Social Welfare | amusement park ride | Recreation | politician | Historical figures: politicians |
| | health | Health | amusement park ride | Recreation | american civil war | Historical figures: politicians |
| | health | Health | amusement park ride | Recreation | lawyer and politician | Historical figures: other |
| | health | Health | amusement park ride | Recreation | historical newspaper | Journalism and newspapers |
| Least suitable | public land | Public Lands | warship and naval unit | Armies and military units | classical greek poetry | Poetry |
| | public land | Public Lands | warship and naval unit | Armies and military units | hinduism | Religious doctrines, teachings, texts, events, and symbols |
| | public land | Environment | warship and naval unit | Military people | hinduism | Religious doctrines, teachings, texts, events, and symbols |
| | indigenous affair | Government Operations | warship and naval unit | Military people | philosophy | Philosophical doctrines, teachings, texts, events, and symbols |
| | indigenous affair | Government Operations | war poetry | Language and literature | philosophy | Philosophical doctrines, teachings, texts, events, and symbols |

Table 6: Assigned LLM labels and ground-truth labels for a given topic from the most and the least suitable cluster configuration according to our algorithm. The purity is higher in the most suitable configuration for LLM labels and ground-truth labels.