# OpenReview forum: "Revisiting Automated Topic Model Evaluation with Large Language Models"
_EMNLP/2023/Conference — EMNLP 2023 Main_

### Official Review · Reviewer_2esg · 2023-08-09

**Soundness:** 4

**Excitement:**

3: Ambivalent: It has merits (e.g., it reports state-of-the-art results, the idea is nice), but there are key weaknesses (e.g., it describes incremental work), and it can significantly benefit from another round of revision. However, I won't object to accepting it if my co-reviewers champion it.

**Paper Topic And Main Contributions:**

The authors proposed to use LLM to evaluate the output of the topic model. The authors have made several claims 1.LLMs can accurately judge topic coherence, 2.LLMs can assist in automatically determining reasonable numbers of topics.

**Questions For The Authors:**

Large language models might focus on common language patterns rather than the specific nuances of certain topics. How do you solve the Overfitting issues  to General Language Patterns in broad topics such as  politics or healthcare?

**Reasons To Accept:**

Automated Topic Model Evaluation with LLM Assistance.

**Reasons To Reject:**

Large language models can be challenging especially if the topics requires domain expertise. Relying on an opaque model to evaluate another model's output might introduce additional uncertainty and lack of transparency.

**Reproducibility:**

4: Could mostly reproduce the results, but there may be some variation because of sample variance or minor variations in their interpretation of the protocol or method.

**Reviewer Confidence:**

4: Quite sure. I tried to check the important points carefully. It's unlikely, though conceivable, that I missed something that should affect my ratings.

---

> ### Author Rebuttal · Authors · 2023-08-28
>
> Thanks a lot for the helpful review which is greatly appreciated!
>
> 1. We agree with your comment about “Relying on an opaque model to evaluate another model's output might introduce additional uncertainty and lack of transparency.” We already have a dedicated section about this in the limitation section of our paper (L321, p. 5). In the final version, we will include this discussion in the main paper because it is important and timely. At the same time, we think there is value in exploring whether and how we can use large language models to tackle long-standing issues in NLP, such as automated evaluation of topic models. This paper contributes to such efforts.
> 2. We explore the overfitting issue in future work where we will explore datasets requiring more in-depth domain knowledge and using large language models specialised in these domains (e.g., the biomedical or legal domain). It is true that there could be an interaction between the evaluating models’ primary domain and its performance on evaluating texts outside of this domain (similarly to quality estimation [1]). However, since this concerns a much larger research question than the scope of this paper, we leave it to future investigations, which are enabled by our present work.
>
> [1] Sharami, Javad Pourmostafa Roshan, et al. ["Tailoring Domain Adaptation for Machine Translation Quality Estimation."](https://arxiv.org/abs/2304.08891) arXiv preprint arXiv:2304.08891 (2023).

---

### Official Review · Reviewer_aBQ5 · 2023-08-09

**Soundness:** 4

**Excitement:**

4: Strong: This paper deepens the understanding of some phenomenon or lowers the barriers to an existing research direction.

**Paper Topic And Main Contributions:**

The paper tackles the problem of automatically evaluating topic models by leveraging LLMs.
In addition, the authors also attempt to use KKNs to automatically determine the optimal number of topics.
The authors find that LLMs can indeed accurately assess topic coherence, and "can assist" in the second task.


**Reasons To Accept:**

- Extensive evaluation
- Interesting read, the proposed approach can be relevant for future research and practitioners

**Reasons To Reject:**

- The paper is hard to read in some parts, see examples below.

**Reproducibility:**

4: Could mostly reproduce the results, but there may be some variation because of sample variance or minor variations in their interpretation of the protocol or method.

**Reviewer Confidence:**

2: Willing to defend my evaluation, but it is fairly likely that I missed some details, didn't understand some central points, or can't be sure about the novelty of the work.

**Typos Grammar Style And Presentation Improvements:**

- I'd remove the mathematical notations and definitions from the first paragraph of the intro, to make it more readable and appealing.
- I'd specify the correlation type measured in Figure 1.

---

> ### Author Rebuttal · Authors · 2023-08-28
>
> Thank you for the review which is greatly appreciated!
>
> 1. We have incorporated your and other suggestions regarding the paper writing.
> 2. In all cases we use Spearman correlation because we are interested in the *ranking* (evaluation) of outputs instead of the absolute values. We specified this in the final version.

---

### Official Review · Reviewer_sobM · 2023-08-10

**Typos Grammar Style And Presentation Improvements:** This paper is well written.
**Soundness:** 4

**Excitement:**

4: Strong: This paper deepens the understanding of some phenomenon or lowers the barriers to an existing research direction.

**Paper Topic And Main Contributions:**

- This paper delves into the feasibility of utilizing LLM for automated topic model evaluation, taking into account prior efforts in evaluation metrics like NPMI, as well as previous research indicating that NPMI does not exhibit a strong correlation with human evaluation.

- The paper further explores the utilization of LLM to automatically determine the ideal number of topics.


**Questions For The Authors:**

- Can you give a brief explanation of how to retrieve the 10 most probable documents in the text labeling algorithm for investigating the number of topics?


**Reasons To Accept:**

- Achieving both objectives of evaluating topic models and determining the optimal number of topics, this paper stands as the first to employ LLM.

- The paper presents a substantial body of insights that can greatly assist the community in leveraging LLM for these specific objectives.


**Reasons To Reject:**

- It would enhance the discussion if the authors acknowledged previous endeavors in nonparametric Bayesian models designed to estimate the number of topics and estimate latent topics.


**Reproducibility:**

3: Could reproduce the results with some difficulty. The settings of parameters are underspecified or subjectively determined; the training/evaluation data are not widely available.

**Reviewer Confidence:**

3: Pretty sure, but there's a chance I missed something. Although I have a good feel for this area in general, I did not carefully check the paper's details, e.g., the math, experimental design, or novelty.

---

> ### Author Rebuttal · Authors · 2023-08-28
>
> Thanks a lot for the helpful review which is greatly appreciated!
>
> 1. In the revised version, we will include a short discussion in the related work section about previous endeavors in nonparametric Bayesian models for estimating the number of topics [e.g. 1].
> 2. The 10 most probable documents are the documents with the highest estimated probability of belonging to a document. LDA estimates a $k$-length distribution over topics $\theta_d$ for each document, which we compose into a $N \times K$ matrix for all $N$ documents. To get the most probable documents per topic, we take an argsort over each matrix column. We will make this more explicit in the final version of the paper. More informally, we retrieve the top ten documents having assigned the highest probability for a given topic by the topic model.
>
> [1] Teh, Yee Whye et al. [“Hierarchical Dirichlet Processes.”](http://www.gatsby.ucl.ac.uk/~ywteh/research/npbayes/jasa2006-print.pdf) Journal of the American Statistical Association 101 (2006): 1566 - 1581.

---

### Meta-Review · Area_Chair_9Uz2 · 2023-09-18

**Recommendation:** 5

**Metareview:**

The reviewers all found this paper sound, with extensive evaluation and a solid experimental setup, and most found it exciting. They noted that the insights from the paper will be beneficial to the community and that the paper is well-grounded in the existing literature on the topic while building upon it. The main reasons to reject had to do with style, formatting, and discussions that could be expanded upon, rather than things like the experimental setup or conclusions being made. One drawback brought up related to using LLMs to evaluate topics in cases where domain expertise is needed, is valid, but is already addressed somewhat by the limitations section. The authors also commit to expanding on this. I do not see this as a major methodological flaw (and I believe the reviewer agrees, given their scores) but rather something to be discussed more in the paper, if accepted.

---

### Decision · Program_Chairs · 2023-10-07

**Decision:**

Accept-Main

**Comment:**

The reviewers all found this paper sound, with extensive evaluation and a solid experimental setup, and most found it exciting. They noted that the insights from the paper will be beneficial to the community and that the paper is well-grounded in the existing literature on the topic while building upon it. The main reasons to reject had to do with style, formatting, and discussions that could be expanded upon, rather than things like the experimental setup or conclusions being made. One drawback brought up related to using LLMs to evaluate topics in cases where domain expertise is needed, is valid, but is already addressed somewhat by the limitations section. The authors also commit to expanding on this. I do not see this as a major methodological flaw (and I believe the reviewer agrees, given their scores) but rather something to be discussed more in the paper, if accepted.